# PEG-8 Laurate Fermentation of *Staphylococcus epidermidis* Reduces the Required Dose of Clindamycin Against *Cutibacterium acnes*

**DOI:** 10.3390/ijms21145103

**Published:** 2020-07-19

**Authors:** Shinta Marito, Sunita Keshari, Chun-Ming Huang

**Affiliations:** 1Department of Biomedical Sciences and Engineering, National Central University, Taoyuan 32001, Taiwan; shintasimbolon53@yahoo.com; 2Department of Life Sciences, National Central University, Taoyuan 32001, Taiwan; sunitakeshari827@gmail.com

**Keywords:** adjuvant, *C. acnes*, clindamycin, PEG-8 Laurate, *S. epidermidis*

## Abstract

The probiotic activity of skin *Staphylococcus epidermidis* (*S. epidermidis*) bacteria can elicit diverse biological functions via the fermentation of various carbon sources. Here, we found that polyethylene glycol (PEG)-8 Laurate, a carbon-rich molecule, can selectively induce the fermentation of *S. epidermidis*, not *Cutibacterium acnes* (*C. acnes*), a bacterium associated with acne vulgaris. The PEG-8 Laurate fermentation of *S. epidermidis* remarkably diminished the growth of *C. acnes* and the *C. acnes*-induced production of pro-inflammatory macrophage-inflammatory protein 2 (MIP-2) cytokines in mice. Fermentation media enhanced the anti-*C. acnes* activity of a low dose (0.1%) clindamycin, a prescription antibiotic commonly used to treat acne vulgaris, in terms of the suppression of *C. acnes* colonization and MIP-2 production. Furthermore, PEG-8 Laurate fermentation of *S. epidermidis* boosted the activity of 0.1% clindamycin to reduce the sizes of *C. acnes* colonies. Our results demonstrated, for the first time, that the PEG-8 Laurate fermentation of *S. epidermidis* displayed the adjuvant effect on promoting the efficacy of low-dose clindamycin against *C. acnes*. Targeting *C. acnes* by lowering the required doses of antibiotics may avoid the risk of creating drug-resistant *C. acnes* and maintain the bacterial homeostasis in the skin microbiome, leading to a novel modality for the antibiotic treatment of acne vulgaris.

## 1. Introduction

The skin is the largest organ, and also home to an ecosystem of a diverse milieu of aerobic and anaerobic microorganisms, most of which are harmless or even beneficial to their host. Microorganisms in the skin microbiome play a chief role in educating the immune system by modulating the cutaneous innate and adaptive immune system [1]. Dysbiosis by microbial imbalance can largely influence its colonization niche by altering the functions of innate and adaptive immunity [2]. A robust evidence for a dysbiotic microbiome is well-documented in inflammatory skin conditions, such as psoriasis, where patients exhibit substantial increases in *Staphylococcus* species on lesional skin. A temporal shift in the composition of the skin microbiome was characterized by significant increases in *Staphylococcus* levels in the skin of atopic dermatitis (AD) patients [3]. Overgrowth of skin commensal *Cutibacterium acnes* (*C. acnes*) (formally named as *Propionibacterium acnes*) has been linked to the cause of inflammatory acne vulgaris, which affects an estimated 80% of Americans. About 94–95% of the pubertal population, 20–40% of adults and <25% of women suffered from acne vulgaris [4]. The study evidenced that *C. acnes* may activate Toll-like receptor (TLR)-2 and TLR-4 on membranes of inflammatory cells releasing inflammatory cytokines, including tumor necrosis factor (TNF)-α and interleukin (IL)-1, IL-8, and IL-12 [5].

Bacterial interference has been found in commensal bacteria, which mediated fermentation to prevent the colonization of pathogens on the host. For example, short-chain fatty acids (SCFAs) or ethanol produced from microbial fermentation of sugars on and inside fruits inhibited the growth of bacterial competitors within ripe fruits [6]. Therapeutic application of bacterial interference by active colonization using *Staphylococcus epidermidis* (*S. epidermidis*), a commensal bacterium in humans, was successful in counteracting the infection of *Staphylococcus aureus* (*S. aureus*) [7,8,9]. Published results from our lab showed that commensal *S. epidermidis* in the skin can ferment glycerol, sucrose or polyethylene glycol (PEG) to produce SCFAs, such as acetic, butyric, lactic, and succinic acids, and successfully inhibited growth of opportunistic *C. acnes* [10,11,12,13]. Although antibiotics have been a common treatment for acne vulgaris for many years, they exert selection pressure on non-target bacteria in skin, running a risk of developing antibiotic-resistant *C. acnes* [14,15]. The incidence of *C. acnes* antibiotic resistance has increased from 20% to 72.5% in 1995, showing that widespread resistance has become a major dermatological issue [16,17]. Major factors, like the chronicity of the disease, antimicrobial administration route, effective penetration in the skin, the duration of therapy, poor treatment compliance, and easy access to therapeutic agents without medical supervision have contributed to the development of antibiotic resistance *C. acnes.* Further, antibiotic-resistant *C. acnes* may also spread from patients to close contacts and is found to be associated with a poor treatment outcome [18,19,20,21]. To reduce the dose of antibiotic, the combination of antibiotic with its adjuvant may be a better strategy to suppress the emergence of resistance and rescue the activity of existing drugs, with little or no antibiotic activity themselves [22]. Poly ethylene glycol (PEG) are most widely used stealth polymer in drug delivery and cosmetics, due to their easy solubility and higher viscosity [23]. PEG has The International Nomenclature of Cosmetic Ingredients (INCI) name, and is thus regulated as safe for use in cosmetics. Moreover, PEGs have a wide variety of PEG-derived mixtures due to their readily linkable terminal primary hydroxyl groups in combination with many possible compounds or complexes, such as ethers or fatty acids [24]. A study revealed that α-PEG Abs was elicited by immunization with PEG coated or PEGylated proteins. However, a little or no immunogenicity was detected by administration with PEG alone suggesting induction of PEG-specific immunity can occur in the absence of adjuvants [25,26]. Moreover, research investigated PEG as a safe adjuvant for HIV and hepatitis B virus DNA vaccines [26,27].

In this study, the polyoxyethylene glycol 400 monolaurate labeled as PEG-8 Laurate (C_28_H_56_O_10_) by INCI was used as a carbon source to induce the fermentation of *S. epidermidis.* We found that PEG-8 Laurate fermentation of *S. epidermidis* can lower the required dose of clindamycin against *C. acnes*, demonstrating the adjuvant effect of PEG-8 Laurate on the anti-*C. acnes* activity of clindamycin.

## 2. Results

### 2.1. PEG-8 Laurate as a Selective Carbon Source for Fermentation of S. epidermidis

To investigate whether *S. epidermidis* ATCC 12228 and *C. acnes* ATCC 6919 can ferment PEG-8 Laurate, bacteria (10^5^ CFU) were incubated with and without 2% PEG-8 Laurate in rich media containing phenol red for 12 h. Rich media with PEG-8 Laurate alone or bacteria alone served as controls. 2% PEG-8 Laurate did not influence the bacterial growth during a 12 h incubation (Appendix A). The color of phenol red in rich media containing *S. epidermidis*, but not *C. acnes*, in the presence of PEG-8 Laurate changed from red to yellow after 12 h incubation, indicating that PEG-8 Laurate is a selective carbon source for the fermentation of *S. epidermidis* (Figure 1a,c). The color change of phenol red was quantified by measuring the optical density at 560 nm (OD_560_) (Figure 1b,d).

### 2.2. PEG-8 Laurate Fermentation of *S. epidermidis* Against C. acnes In Vivo

To evaluate whether PEG-8 Laurate fermentation of *S. epidermidis* can influence the growth of *C. acnes* in vivo, *S. epidermidis* (10^7^ CFU) and *C. acnes* (10^7^ CFU), with or without 2% PEG-8 Laurate, were co-injected intradermally into ear of Institute for Cancer Research (ICR) mice for five days. In Figure 2, compared to those mice injected with two bacteria plus H_2_O, mice injected with two bacteria plus PEG-8 Laurate showed a decrease in both ear redness and thickness (Figure 2a,b). The number (CFU/mL) of *C. acnes* in mouse ears injected with two bacteria in the presence of PEG-8 Laurate was approximately one log_10_ lower than that in ears injected with two bacteria in the absence of PEG-8 Laurate (Figure 2c). Since acne vulgaris is an inflammatory skin disorder [5], we next determined whether PEG-8 Laurate fermentation of *S. epidermidis* can ameliorate the production of *C. acnes*-induced pro-inflammatory cytokines. The mouse ears were excised and homogenized five days after bacterial injection. The level of macrophage-inflammatory protein-2 (MIP-2), a murine counterpart of human IL-8 [13], in ear homogenates was measured by an enzyme-linked immunosorbent assay (ELISA). As shown in Figure 2d, the level of MIP-2 in the ear injected with two bacteria plus PEG-8 Laurate was significantly lower than that in the ear injected with two bacteria and H_2_O (Figure 2d). Furthermore, in terms of the number of *C. acnes* and level of MIP-2, we found that there were no differences in mouse ears injected, with or without 2% PEG-8 Laurate, alone in the absence of bacteria, demonstrating that PEG-8 Laurate also have no impact on *C. acnes* growth and MIP-2 production (Appendix A). The results in Figure 2 suggested that *S. epidermidis* mediated PEG-8 Laurate for fermentation, to mitigate the growth of *C. acnes* and inflammation.

### 2.3. Reduction of the Effective Dose of Clindamycin Against C. acnes by Fermentation Media

Clindamycin is one of the common antibiotics for acne vulgaris [28]. However, the antibiotic single-agent therapy can result in the rapid development of clinically significant antibiotic resistance [28]. Several carbohydrates, for example sucrose, have been added into antibiotic (amoxicillin) formulae as ingredients [29]. An adjuvant is a pharmacological or immunological agent that modifies the effect of a drug or vaccine [29,30]. To evaluate the effect of PEG-8 Laurate fermentation on the efficacies of anti-*C. acnes* antibiotics, we injected the ears of ICR mice intradermally with *C. acnes* (10^7^ CFU) with 0.1%, 1% clindamycin or H_2_O, along with media collected from the culture of S. *epidermidis* in the presence or absence of 2% PEG-8 Laurate. Six days after injection, ears were excised and homogenized for the quantification of the number of *C. acnes* and the level of MIP-2. As shown in Figure 3, the injection of 1%, but not 0.1% clindamycin, along with culture media of *S. epidermidis*, caused a remarkable decrease in the number of *C. acnes* and level of MIP-2. However, the 0.1% clindamycin became effective in terms of reduction of the number of *C. acnes* (Figure 3b) and level of MIP-2 (Figure 3c), when it was co-injected with fermentation media of *S. epidermidis* in the presence of PEG-8 Laurate. The number of *C. acnes* in ears injected with *C. acnes*/0.1% clindamycin and media of *S. epidermidis* plus PEG-8 Laurate was one log_10_ lower than that in ears injected with *C. acnes*/0.1% clindamycin and media of *S. epidermidis*. The levels of MIP-2 in ears injected with *C. acnes*/0.1% clindamycin and media of *S. epidermidis* with or without PEG-8 Laurate are 3469 ± 26.61 and 7685 ± 561.5 ng/mL, respectively. These results indicated that molecules in fermentation media could enhance the efficacy of clindamycin against *C. acnes*.

### 2.4. Enhancement of Low Dose of Clindamycin for Inhibition of the Growth of C. acnes Colonies by PEG-8 Laurate Fermentation

To confirm that the PEG-8 Laurate fermentation of *S. epidermidis* can augment the activity of low dose of clindamycin against *C. acnes*, *C. acnes* in the presence of 0.1% or 1% clindamycin was added onto the top of a homogeneous lawn of *S. epidermidis*, with or without 2% PEG-8 Laurate in agar plates. As shown in Figure 4, the addition of 0.1% clindamycin with *C. acnes* onto the *S. epidermidis* lawn without PEG-8 Laurate can significantly reduce the size of a *C. acnes* colony. The *C. acnes* colony was nearly completely eliminated when 1% clindamycin with *C. acnes* was added onto the *S. epidermidis* lawn without PEG-8 Laurate. Moreover, 2% PEG-8 Laurate was supplemented into the *S. epidermidis* lawn to induce the *S. epidermidis* fermentation. An inhibition zone was observed at the border of a *C. acnes* colony on the *S. epidermidis* lawn with 2% PEG-8 Laurate (Figure 4a), illustrating the bacterial interference by PEG-8 Laurate fermentation of *S. epidermidis*. Supplementation of PEG-8 Laurate into the *S. epidermidis* lawn considerably enhanced the ability of 0.1% clindamycin to eradicate the *C. acnes* colony (Figure 4b). The result demonstrated that PEG-8 Laurate fermentation of *S. epidermidis* could boost the low dose of clindamycin against *C. acnes*.

## 3. Discussion

Different bacteria express distinct enzymes that metabolize carbon sources during fermentation [31]. Previous results from our laboratory have shown that *S. epidermidis*, but not *C. acnes*, can ferment sucrose to produce SCFAs [13]. Here, we found that PEG-8 Laurate could selectively induce fermentation of *S. epidermidis*, but not *C. acnes* (Figure 1). Thus, as carbon sources, sucrose and PEG-8 Laurate may function as prebiotics, which can specifically provoke the fermentation of *S. epidermidis* for the production of SCFAs. It has been reported that SCFAs such as succinic acid and acetic acid can effectively suppress the growth of *C. acnes* [13]. The fact that the production of SCFAs with anti-*C. acnes* activities in fermentation media of *S. epidermidis* may well explain why PEG-8 Laurate can lower the required dose of clindamycin for killing *C. acnes*. Our previous results have identified many SCFAs produced by PEG fermentation of *S. epidermidis* via gas chromatography mass spectrometry (GC-MS) analysis [12]. Future work will include the identification of SCFAs in the media of PEG-8 Laurate fermentation of *S. epidermidis*. Lauric acid or its derivative elicits antimicrobial activity against *C. acnes* [32]. However, our data demonstrated that 2% PEG-8 Laurate itself, as a lauric acid derivative, did not influence the growth of *C. acnes* (Appendix A). PEG-8 Laurate has been used in cosmetics and beauty products as a surfactant and emulsifying agent. Our study demonstrated, for the first time, that skin *S. epidermidis* bacteria can ferment PEG-8 Laurate to counteract *C. acnes*.

Adjuvants for antibiotics include small molecules, which can enhance the effects of antibiotics without killing the bacteria directly by themselves. Therapy using antibiotic adjuvants may be a potent way to suppress the formerly susceptible bacteria that have acquired resistance [33]. Furthermore, β-Lactamase inhibitors, in combination with β-lactam antibiotics, were found to be successful for the treatment of bacterial infections [34,35,36]. The successful inhibition of *Acinetobacter baumannii,* an extensively drug-resistant strain causing 50% mortality rates when causing ventilator-associated pneumonia, has been achieved by 2-aminoimidazole-based compounds, which disrupt two-component signaling, and anthracyclines, that potentiate the activity of rifampin and linezolid [37,38]. Antibiotic resistance is a major hurdle in the treatment of infectious diseases. Reduction of the *frequency* and intensity of *antibiotic* consumption may be a way to prevent the development of antibiotic resistant strains. Any product, pathway, or phenotype that contributes to decreased antibiotic susceptibility represents a potential adjuvant target, many of which have yet to be fully probed. Here, we demonstrated that the PEG-8 Laurate fermentation of *S. epidermidis* can reduce the effective dose of clindamycin for killing *C. acnes* (Figure 3 and Figure 4). PEG-8 Laurate, thus, functions as an adjuvant to promote the anti-*C. acnes* efficacy of clindamycin, by stimulating the probiotic activity of *S. epidermidis*. Detection of *C. acnes* induced inflammatory cytokines provides a validation on effective inhibition on *C. acnes* growth. CXCL1/keratinocyte-derived chemokine (KC), CXCL2/MIP2 and CXCL5-6/lipopolysaccharide-induced CXC chemokine (LIX) were regarded as functional homologues of IL-8, where CXCL1, CXCL2 and CXCL6 are identified as candidates associated with acne pathogenesis in humans [39,40]. Additionally, multiple studies in the mouse acne model showed a remarkable increase in the production of neutrophils, macrophages and MIP-2 upon *C. acnes* administration into mice [41,42,43]. In our study, we investigated that PEG-8 Laurate fermentation of *S. epidermidis* could potentially ameliorate the effective dose of clindamycin to reduce *C. acnes*-induced MIP-2 production (Figure 3). Future studies involving the detection of CXCL1 and CXCL6 in murine model of acne may provide additional information to correlate these inflammatory mediators with acne severity.

Over the years, an increased antibiotic resistance to *C. acnes* was detected to tetracycline and clindamycin. The prevalent rates of *C. acnes* with erythromycin or tetracycline resistance ranges from 5% to 26.4%, however 45% to 91% clindamycin resistance was detected in acne vulgaris [44]. Although at earlier phases, the efficacy of in acne treatment has been shown to be sustained, there was still an increase in the number of resistant bacteria in samples from patients using 1% clindamycin alone [45]. Although the spectrum of antimicrobial activity of clindamycin includes staphylococci, streptococci and pneumococci and most anaerobic bacteria [46], our result showed that 1% clindamycin can efficiently kill anaerobic *C. acnes*, but not *S. epidermidis*, a facultative anaerobic bacterium (Appendix A). The 0.1% clindamycin did not influence the interference of *S. epidermidis* with *C. acnes* in the absence of PEG-8 Laurate (Figure 3), although it can significantly reduce the size of a *C. acnes* colony in a *S. epidermidis* lawn (Figure 4). The supplementation of PEG-8 Laurate into a *S. epidermidis* lawn significantly enhanced the anti-*C. acnes* activity of 0.1% clindamycin. Clindamycin acts by inhibiting the protein synthesis of bacteria at the level of the 50S ribosome. PEG-8 Laurate initiated *S. epidermidis* fermentation and may induce the production of SCFAs, which can lower the intracellular pH values of bacteria [11], leading the lysis of *C. acnes*. The INCI name of PEG-8 and its efficacy as a potential prebiotic made it safer for extensive usage in cosmetics and drugs [12,24]. In this study, *C. acnes* ATCC 6919, a phylotype IA1, was used to demonstrate the adjuvant effect of PEG-8 Laurate fermentation of *S. epidermidis* against *C. acnes*. The phylotype IA1 of *C. acnes* was predominantly associated with acne vulgaris. The phylotype IA1 of *C. acnes* displayed clindamycin sensitivity, whereas phylotype IA2 frequently associated with severe acne vulgaris and phylotype IB associated with commensal *C. acnes* exhibited high clindamycin resistance [47]. Future studies will thus investigate the effect of PEG-8 Laurate fermentation of *S. epidermidis* on clindamycin against phylotypes IA2 and IB of *C. acnes*.

In summary, a “*drug repurposing*” strategy unveils many compounds that have the unexpected potential to treat different diseases. Although PEG-8 Laurate is currently used as an emulsifying agent in cosmetic products, we repurpose it as a prebiotic, which can induce the fermentation of skin probiotic *S. epidermidis*. PEG-8 Laurate fermentation of *S. epidermidis* reduced the therapeutically effective dose of clindamycin, demonstrating the potential of PEG-8 Laurate as an antibiotic adjuvant for treatments of acne vulgaris and other skin infections.

## 4. Materials and Methods

### 4.1. Ethics Statement

This research was carried out in strict accordance with an approved Institutional Animal Care and Use Committee (IACUC) protocol at National Central University (NCU), Taiwan (NCU-106-016, 19 December 2017). ICR mice (8–9-week-old females) obtained from National Laboratory Animal Centre, Taipei, Taiwan, were sacrificed in dry ice (solid carbon dioxide) in a closed box.

### 4.2. Bacterial Culture

*S. epidermidis* ATCC 12228 bacteria were cultured in tryptic soy broth (TSB) (Sigma, St. Louis, MO, USA) overnight at 37 °C. *C. acnes* ATCC 6919 bacteria were cultured on Reinforced Clostridium Medium (RCM) (ThermoFisher Scientific, Waltham, MA, USA), and cultured under anaerobic conditions using a Gas-Pak (BD Biosciences, San Jose, CA, USA) at 37 °C, until the logarithmic growth phase. The cultures were diluted at 1:100 and OD was measured at 600 nm (OD_600_) = 1.0. Bacterial pellets were harvested by centrifugation at 5000× *g* for 10 min, washed with phosphate-buffered saline (PBS), and suspended in PBS for further experiments.

### 4.3. Fermentation of Bacteria

*S. epidermidis* or *C. acnes* (10^5^ CFU/mL) was incubated in 10 mL rich media (10 g/L yeast extract (Biokar Diagnostics, Beauvais, France), 3 g/L TSB, 2.5 g/L K_2_HPO_4_, and 1.5 g/L KH_2_PO_4_), in the presence and absence of 20 g/L PEG-8 Laurate (Taiwan NJC Corporation), under aerobic/anaerobic conditions, at 37 °C overnight shaking at 200 rpm. In some experiments, 20 g/L PEG-8 Laurate was added into media without bacteria. The 0.002% (w/v) phenol red (Sigma) in rich media was served as a fermentation indicator. A color change from red to yellow indicated the occurrence of bacterial fermentation, and it was detected by measuring OD at 560 nm (OD_560_).

### 4.4. PEG-8 Laurate Fermentation of S. epidermidis Against C. acnes In Vivo

ICR mice were anesthetized by isoflurane (Panion and BF Biotech Inc., Taiwan). Three mice per group were used in each experiment. The ears of ICR mice were injected intradermally with *C. acnes* ATCC 6919 (10^7^ CFU) and *S. epidermidis* ATCC 12228 (10^7^ CFU), with or without 2% PEG-8 Laurate for 5 days. In some experiments, mouse ears were injected intradermally with *C. acnes* (10^7^ CFU) and 0.1 or 1% clindamycin, along with media collected from the culture of *S. epidermidis* (10^7^ CFU), in the presence or absence of 2% PEG-8 Laurate for 6 days. Thickness of mouse ear was measured every day with an electronic digital caliper (Mitutoyo, Kanagawa, Japan). Ear redness or erythema was photographed using a USB digital microscope (GM019-#1, Global Camera Manufacturer Corp, Taipei, Taiwan). Mouse ears were excised and homogenized for detection of MIP-2 and bacterial counts 5 or 6 days after bacterial injection.

### 4.5. Bacterial Loads in Mouse Ears

After excising mouse ears, tissue homogenates were made by a tissue grinder in 200 μL of sterile PBS. CFUs of *C. acnes* in ear homogenates were enumerated by plating serial dilutions (1:10^0^–1:10^5^) of homogenates on *C. acnes* selective agar plate containing RCM and 10µg/mL of furazolidone (PubChem, Rockville Pike, Bethesda, MD, USA). Plates were incubated for 3 days at 37 °C under an anaerobic condition, using a Gas-Pak.

### 4.6. An Overlay Assay

*S. epidermidis* ATCC 12228 (10^8^ CFU) in 300 µL PBS was poured into the plates, containing 2% RCM agar, with or without 2% PEG -8 Laurate, to produce a homogeneous lawn of *S. epidermidis*. Furthermore, *C. acnes* ATCC6919 (10^6^ CFU), with or without 0.1 and 1% clindamycin in 10 µL water, was added on the top of the lawn of *S. epidermidis* before incubation at 37 °C for 5 days. Addition of *C. acnes* without clindamycin was included as a control.

### 4.7. ELISA

Mouse ears were excised and homogenized with T-PER^TM^ Tissue Protein Extraction Reagent (ThermoFisher Scientific), supplemented with ethylenediaminetetraacetic acid (EDTA)-free protease inhibitor cocktail (Sigma). Ear homogenates were centrifuged at 15,000 rpm for 30 min at 4 °C and supernatant was collected to measure MIP-2 concentration by sandwich ELISA, using a Quantikine mouse MIP-2 set (R&D Systems, Minneapolis, MN, USA).

### 4.8. Statistical Analysis

Data analysis was performed by unpaired Student’s t-test or by 1-way ANOVA using Prism software (GraphPad Software, La Jolla, CA, USA). The bacterial inhibition zone in the overlay assay was measured using ImageJ software (National Institutes of Health, Bethesda, MD, USA). The *p*-values of <0.05 (*), <0.01 (**), and <0.001 (***) were considered as significant. The mean ± SD for at least three independent experiments was calculated.

## Figures and Tables

**Figure 1 ijms-21-05103-f001:**
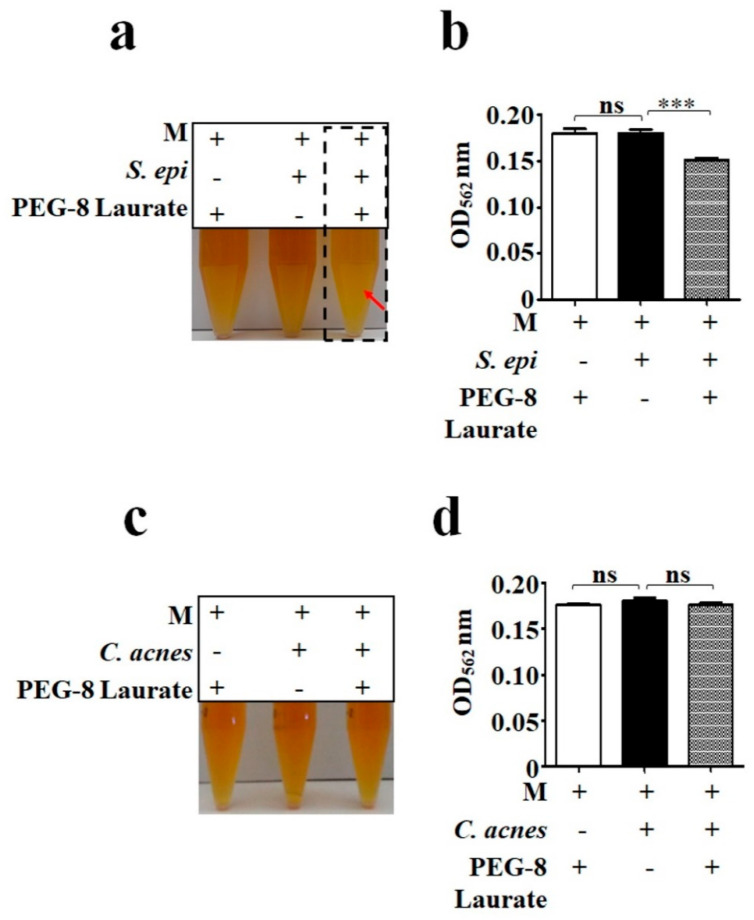
PEG-8 Laurate as a carbon source for fermentation of *S. epidermidis,* but not *C. acnes.* (**a**,**b**) *S. epidermidis* (*S. epi*) or (**c**,**d**) *C. acnes* (10^5^ CFU/mL) was incubated with and without 2% PEG-8 Laurate in rich media (M) for 12 h. Rich media plus PEG-8 Laurate alone or bacteria alone served as controls. Bacterial fermentation was indicated by the color change of phenol red to yellow (arrow) and quantified by measurement of OD_560_ (**b**,**d**). Results were illustrated as the mean ± standard deviation (SD) of three independent experiments. *** *p* < 0.001 (two-tailed *t*-tests); ns = non-significant.

**Figure 2 ijms-21-05103-f002:**
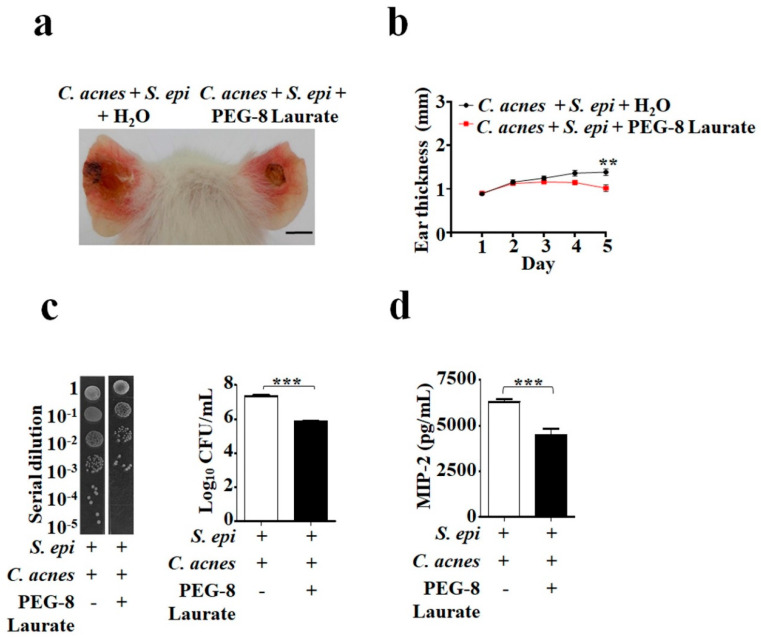
Inhibition of *C. acnes* growth and MIP-2 production by PEG-8 Laurate fermentation of *S. epidermidis* in vivo. The ears of ICR mice were intradermally injected with *C. acnes* (10^7^ CFU) and *S. epidermidis* (*S. epi*), (10^7^ CFU) with 2% PEG-8 Laurate or H_2_O. Ear morphology 5 d after injection (**a**) and changes in ear thickness (mm) (**b**) were shown. Scale bar = 1 mm. (**c**) The CFUs (Log_10_ CFU/mL) were assessed by enumerating plating serial dilution (1:10^0^–1:10^5^) of the ear homogenates on agar plates. (**d**) The levels of MIP-2 cytokines in ear homogenates were measured by ELISA. Data are represented as mean ± SD of three separate experiments, using three mice per group. *** *p* < 0.001 (two-tailed *t*-tests).

**Figure 3 ijms-21-05103-f003:**
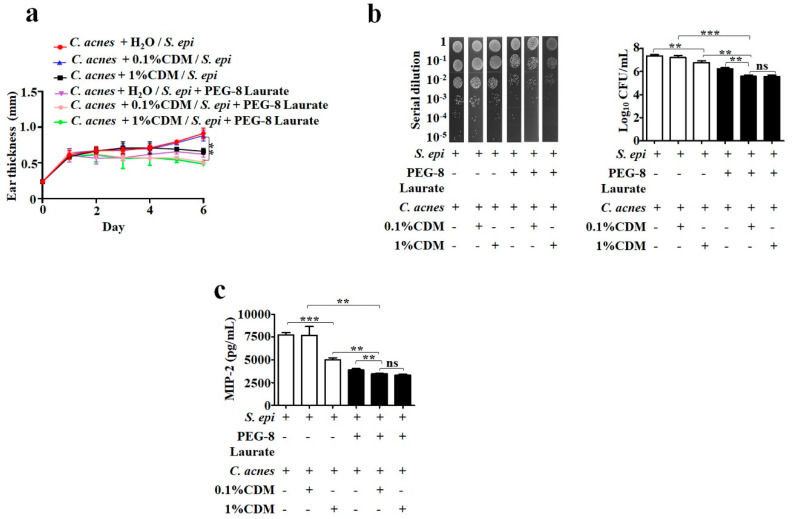
Reduction of the anti-*C. acnes* dose of clindamycin in the presence of fermentation media. The ears of ICR mice were intradermally injected with *C. acnes* (10^7^ CFU), plus 0.1 or 1% of clindamycin (CDM) or H_2_O, along with media collected from culture of *S. epidermidis* (*S. epi*), with or without 2% PEG-8 Laurate. (**a**) Ears thickness (mm) was measured for 6 days after injection. (**b**) The CFUs were counted after plating serial dilution (1:10^0^–1:10^5^) of ear homogenates agar plates. (**c**) The levels of MIP-2 cytokines in ear homogenates were quantified by ELISA. Data are represented as mean ± SD of three independent experiments using three mice per group. ** *p* < 0.05; *** *p* < 0.001 (two-tailed *t*-tests); ns = non-significant.

**Figure 4 ijms-21-05103-f004:**
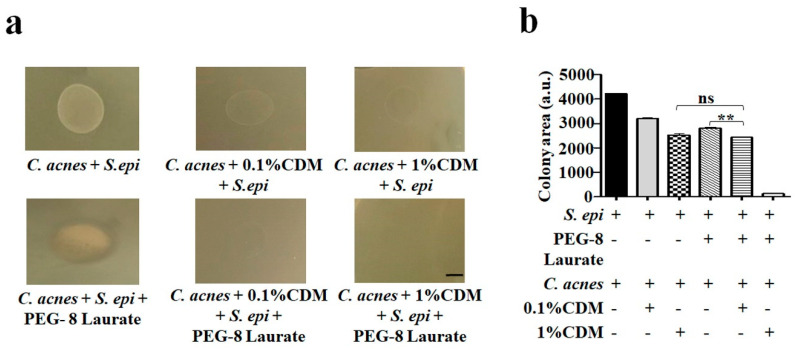
Enhancement of anti-*C. acnes* activity and reduction of required doses of clindamycin by PEG-8 fermentation of *S. epidermidis* in an overlay assay. (**a**) *C. acnes* (10^6^ CFU) in the presence or absence of 1 or 0.1% clindamycin (CDM) was overlaid on top of a lawn of *S. epidermidis* (10^8^ CFU), with or without 2% PEG-8 Laurate. Scale bar = 1 cm (**b**) Graphical presentation of quantification of area (arbitrary unit, a.u.) of a *C. acnes* colony. Results display as mean ± SD in triplicate. ** *p* < 0.05 (two-tailed *t*-tests); ns = non-significant.

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
