# Peer review of "PEG-8 Laurate Fermentation of Staphylococcus epidermidis Reduces the Required Dose of Clindamycin Against Cutibacterium acnes"

_ijms, 2020, doi:10.3390/ijms21145103_

Round 1

Reviewer 1 Report

The manuscript entitled “ “ by et al. intends to demonstrate that PEG-8 Laurate, currently used as an emulsifying agent in cosmetic products, may be employed as a prebiotic inducing the fermentation of skin probiotic S. epidermidis and acting as an antibiotic adjuvant. The manuscript is in the scopus of International Journal of Molecular Sciences. However, in my opinion, the English need to be revised by a native speaker. I also have some major concerns:

  • I do not understand the sense of this sentence in the manuscript “Microorganisms metabolized sugars on and inside fruits to produce fermentation products including short-chain fatty acids (SCFAs) which inhibited the growth of bacterial competitors within ripe fruits”
  • Include reference “The incidence of C. acnes antibiotic resistance has increased from 20% to 72.5% in 1995 showing widespread resistance has become a major dermatological issue.”. Indeed, a more recent reference should be mentioned.
  • Introduction should be improved regarding the advantageous and previous studies on PEG-8 Laurate
  • Why the authors selected 5 days and 6 days (depending on the finality) for the in vivo assays?
  • Figure should have higher resolution
  • Discussion section has to be improved.

Author Response

Reviewer 1

Comment 1

The manuscript entitled “ “ by et al. intends to demonstrate that PEG-8 Laurate,  currently used as an emulsifying agent in cosmetic products, may be employed as a prebiotic inducing the fermentation of skin probiotic S. epidermidis and acting as an antibiotic adjuvant. The manuscript is in the scopus of International Journal of Molecular Sciences. However, in my opinion, the English need to be revised by a native speaker.

 Response 1

 English has been revised.

Comment 2

I do not understand the sense of this sentence in the manuscript “Microorganisms metabolized sugars on and inside fruits to produce fermentation products including short-chain fatty acids (SCFAs) which inhibited the growth of bacterial competitors within ripe fruits” Include reference “The incidence of C. acnes antibiotic resistance has increased from 20% to 72.5% in 1995 showing widespread resistance has become a major dermatological issue.”. Indeed, a more recent reference should be mentioned.

   Response 2

  The sentence “Microorganisms metabolized sugars on and inside fruits to produce fermentation products including short-chain fatty acids (SCFAs) which inhibited the growth of bacterial competitors within ripe fruits” is the example of bacterial interference via fermentation in natural ecosystems.  We have modified this sentence as below and cited reference [6].

For example, short-chain fatty acids (SCFAs) or ethanol produced from microbial fermentation of sugars on and inside fruits inhibited the growth of bacterial competitors within ripe fruits.”

  1. Dudley, R. Ethanol, fruit ripening, and the historical origins of human alcoholism in primate frugivory. Integrative and comparative biology 2004, 44, 315-323.

For, “The incidence of C. acnes antibiotic resistance has increased from 20% to 72.5% in 1995 showing widespread resistance has become a major dermatological issue.”. We have cited 2 recent references [16,17].

  1. Humphrey, S. Antibiotic resistance in acne treatment. Skin Therapy Lett 2012, 17, 1-3.
  2. Williams, H.C.; Dellavalle, R.P.; Garner, S. Acne vulgaris. The Lancet 2012, 379, 361-372.

Comment 3

 Introduction should be improved regarding the advantageous and previous studies on PEG-8 Laurate

 Response 3

     We have added a new paragraph stating the advantageous and previous studies on PEG-8 Laurate as below and 4 references have been cited.

     Poly ethylene glycol (PEG) are most widely used stealth polymer in drug delivery and cosmetics due to their easy solubility, higher viscosity [23]. PEG has The International Nomenclature of Cosmetic Ingredients (INCI) name, thus regulated as safe for use in cosmetics. Moreover, PEGs have a wide variety of PEG-derived mixtures due to their readily linkable terminal primary hydroxyl groups in combination with many possible compounds or complexes such as ethers or fatty acids [24]. Study revealed that α-PEG Abs was elicited by immunization with PEG coated or PEGylated proteins. However, a little or no immunogenicity was detected by administration with PEG alone suggesting induction of PEG-specific immunity can occur in the absence of adjuvants [25,26].

  1. Fruijtier, C. Safety assessment on polyethylene glycols (PEGs) and their derivatives as used in cosmetic products. Toxicology 2005, 214, 1-38, doi:10.1016/j.tox.2005.06.001.
  2. Jang, H.-J.; Shin, C.Y.; Kim, K.-B. Safety Evaluation of Polyethylene Glycol (PEG) Compounds for Cosmetic Use. Toxicol Res 2015, 31, 105-136, doi:10.5487/tr.2015.31.2.105.

      25.Yang, Q.; Lai, S.K. Anti-PEG immunity: emergence, characteristics, and unaddressed questions. Wiley Interdiscip Rev Nanomed Nanobiotechnol 2015, 7, 655-677, doi:10.1002/wnan.1339.

   26.Liu, Y.; Balachandran, Y.L.; Li, D.; Shao, Y.; Jiang, X. Polyvinylpyrrolidone–Poly(ethylene glycol) Modified Silver Nanorods Can Be a Safe, Noncarrier Adjuvant for HIV Vaccine. ACS Nano 2016, 10, 3589-3596, doi:10.1021/acsnano.5b08025.

 Comment 4

Why the authors selected 5 days and 6 days (depending on the finality) for the in vivo assays?

Response 4

In 1st in vivo assay the ears of ICR mice were injected intradermally with C. acnes ATCC 6919 (107 CFU) and S. epidermidis ATCC 12228 (107 CFU) with or without 2% PEG-8 Laurate for 5 days, a time period when the effect of PEG-8 Laurate fermentation of S. epidermidis against C. acnes growth was observed.

In 2nd in vivo assay the ears of ICR mice were injected intradermally with C. acnes (107 CFU) and 0.1 or 1% clindamycin along with media collected from the culture of S. epidermidis (107 CFU) in the presence or absence of 2% PEG-8 Laurate for 6 days, a time period when the effect of dose of clindamycin in the presence of fermentation media from PEG-8 Laurate fermentation of S. epidermidis to inhibit C. acnes growth was detected.

Comment 5

 Figure should have higher resolution

 Response 5

  We have added the figures with enhanced resolution.

 Comment 6

Discussion section has to be improved.

 Response 6

 We have revised the discussion section and underlined all the changes in the manuscript.

Reviewer 2 Report

The paper by Shinta Marito and colleagues is really very interesting and objectively promising also thanks to the renewed interest in the biome as a modulator in the most varied pathological areas.

However, many aspects related to the results should be expanded.

In figure 1 in the legend it is said that C.acnes was incubated with and without PEG-8 Laurate in rich media, but it is shown in the figure in a non-homogeneous way and all conditions should be shown in all panels a, b, c , d.

Human IL8 does not have a real linearly identifiable murine counterpart. CXCL2 / MIP2 has many similar functions and is counted as one of the chemokines with a comparable function. However, CXCL1 / KC and CXCL5-6 / Lix also do the same. In general it would be better to look at all the chemokines or rephrase while keeping less confident that human IL8 behaves in the exact same way.

Figure 2a shows the image of the mouse with the double inoculation +/- PEG-8 Laurate, but the mouse ctr and with the single strains is not shown.

In figure 2b the symbols that should identify the two treatments are very similar and make reading the graph not immediate and difficult.

Although not univocal, but the levels of MIP-2 at baseline on the murine strain must be indicated, as well as those of the mice treated individually with the two strains (figure 2d).

In figure S2a the legend is completely missing and also in this case the symbols are very similar.

In figure S2b, c the PEG-8 Laurate alone is missing

In general, the graphic appearance of the figures is very poor as they appear stretched and rearranged after processing; the size and style of the characters is random, resulting too small or too large depending on the figure examined. In general they should be standardized with a single working hand and favoring the readability of the text and above all the graphic information.

Brand and model should be indicated on line 262 for the USB digital microscope

For the statistical analysis described starting from line 282, it should be specified whether the calculation was done manually or with specific software. In the first case the formula (s) should be shown; in the second the program used.

Author Response

Reviewer 2

The paper by Shinta Marito and colleagues is really very interesting and objectively promising also thanks to the renewed interest in the biome as a modulator in the most varied pathological areas. However, many aspects related to the results should be expanded.

 Comment 1

In figure 1 in the legend it is said that C. acnes was incubated with and without PEG-8 Laurate in rich media, but it is shown in the figure in a non-homogeneous way and all conditions should be shown in all panels a, b, c , d.

 Response 1

 We have corrected all the treatment conditions in all panels (a, b, c, d) of figure 1

Comment 2

Human IL8 does not have a real linearly identifiable murine counterpart. CXCL2 / MIP2 has many similar functions and is counted as one of the chemokines with a comparable function. However, CXCL1 / KC and CXCL5-6 / Lix also do the same. In general it would be better to look at all the chemokines or rephrase while keeping less confident that human IL8 behaves in the exact same way.

 Response 2

Detection of C. acnes induced inflammatory cytokines provides a validation on effective inhibition on C. acnes growth. CXCL1 / keratinocyte-derived chemokine (KC), CXCL2 / MIP2 and CXCL5-6 / lipopolysaccharide-induced CXC chemokine (LIX) regarded as functional homologues of IL-8 where CXCL1, CXCL2 and CXCL6 are identified as candidates associated with acne pathogenesis in human [40,41]. Additionally, multiple studies in mouse acne model showed a remarkable increase in the production of neutrophils, macrophages and MIP-2 upon C. acnes administration into mice [42-44]. In our study, we investigated that PEG-8 Laurate fermentation of S. epidermidis could potentially ameliorate the effective dose of clindamycin to reduce C. acnes-induced MIP-2 production (Figure 3). Future studies involving detection of CXCL1 and CXCL6 in murine model of acne may provide additional information to correlate these inflammatory mediators with acne severity.

  1. Li, X.; Jia, Y.; Wang, S.; Meng, T.; Zhu, M. Identification of Genes and Pathways Associated with Acne Using Integrated Bioinformatics Methods. Dermatology 2019, 235, 445-455.
  2. Kelhala, H.L.; Palatsi, R.; Fyhrquist, N.; Lehtimaki, S.; Vayrynen, J.P.; Kallioinen, M.; Kubin, M.E.; Greco, D.; Tasanen, K.; Alenius, H., et al. IL-17/Th17 pathway is activated in acne lesions. PLoS One 2014, 9, e105238, doi:10.1371/journal.pone.0105238.
  3. Nakatsuji, T.; Shi, Y.; Zhu, W.; Huang, C.P.; Chen, Y.R.; Lee, D.Y.; Smith, J.W.; Zouboulis, C.C.; Gallo, R.L.; Huang, C.M. Bioengineering a humanized acne microenvironment model: proteomics analysis of host responses to Propionibacterium acnes infection in vivo. Proteomics 2008, 8, 3406-3415, doi:10.1002/pmic.200800044.
  4. Achermann, Y.; Goldstein, E.J.; Coenye, T.; Shirtliff, M.E. Propionibacterium acnes: from commensal to opportunistic biofilm-associated implant pathogen. Clin Microbiol Rev 2014, 27, 419-440, doi:10.1128/CMR.00092-13.
  5. Itakura, M.; Tokuda, A.; Kimura, H.; Nagai, S.; Yoneyama, H.; Onai, N.; Ishikawa, S.; Kuriyama, T.; Matsushima, K. Blockade of Secondary Lymphoid Tissue Chemokine Exacerbates <em>Propionibacterium acnes-</em>Induced Acute Lung Inflammation. The Journal of Immunology 2001, 166, 2071, doi:10.4049/jimmunol.166.3.2071.

Comment 3

Figure 2a shows the image of the mouse with the double inoculation +/- PEG-8 Laurate, but the mouse ctr and with the single strains is not shown.

Response 3

We have conducted experiment including mouse control and the figure is shown in Supplementary Fig. 2. (Fig S2)

 Data from experiment using S.epidermidis alone and +/- PEG-8 Laurate was shown in;

 Fig.1 in in vitro assay to validate its fermenting activity using PEG-8 Laurate

  Supplementary Fig 1 (Fig. S1) to validate that PEG-8 Laurate does not kill S.epidermidis

 Data from experiment using C. acnes alone and +/- PEG-8 Laurate was shown in;

     Fig.1 in in vitro assay to validate its non-fermenting activity using PEG-8 Laurate

     Supplementary Fig 1 (Fig. S1) to validate that PEG-8 Laurate does not kill C. acnes

     Supplementary Fig 2 (Fig. S2) to show C. acnes induced MIP-2 and bacteria number.

Comment 4

In figure 2b the symbols that should identify the two treatments are very similar and make reading the graph not immediate and difficult.

Response 4

In figure 2b, we have modified the symbols that identify the treatments.

Comment 5

Although not univocal, but the levels of MIP-2 at baseline on the murine strain must be indicated, as well as those of the mice treated individually with the two strains (figure 2d).

Response 5

The level of MIP-2 in ear of control mice and mice injected with PEG alone has been detected and the new result is shown in supplementary figure 2 (Fig S2).

Comment 6

In figure S2a the legend is completely missing and also in this case the symbols are very similar.

Response 6

We have added the legends in figure S2a and modified the symbols for the treatment groups

Comment 7

In figure S2b, c the PEG-8 Laurate alone is missing

Response 7

We have conducted experiment for the mice group with PEG-8 Laurate alone and the new figure is shown in figure S2b, c  

Comment 8

In general, the graphic appearance of the figures is very poor as they appear stretched and rearranged after processing; the size and style of the characters is random, resulting too small or too large depending on the figure examined. In general they should be standardized with a single working hand and favoring the readability of the text and above all the graphic information.

Response 8

We have added the figures with enhanced resolutions. The size and style of the characters are revised.

Comment 9

Brand and model should be indicated on line 262 for the USB digital microscope

Response 9

We have added the brand and model of USB digital microscope as (GM019-#1, Global Camera Manufacturer Corp, Taipei, Taiwan).

Comment 10

For the statistical analysis described starting from line 282, it should be specified whether the calculation was done manually or with specific software. In the first case the formula (s) should be shown; in the second the program used.

Response 10

Data analysis was performed by unpaired Student’s t-test or by 1-way ANOVA using Prism software (GraphPad Software, La Jolla, CA, USA). The bacterial inhibition zone in the overlay assay was measured using ImageJ software (National Institutes of Health, Bethesda, MD, USA).

Reviewer 3 Report

In this paper, the authors found that PEG-8 Laurate fermentation of S. epidermidis  can lower the required dose of clindamycin against C. acnes, demonstrating the adjuvant effect of  PEG-8 Laurate on anti-C. acnes activity of clindamycin.

Overall the manuscript is well organized and well written.

Author Response

Reviewer 3

In this paper, the authors found that PEG-8 Laurate fermentation of S. epidermidis can lower the required dose of clindamycin against C. acnes, demonstrating the adjuvant effect of PEG-8 Laurate on anti-C. acnes activity of clindamycin. Overall the manuscript is well organized and well written.

Response

We appreciate the thoughtful and insightful comment by the reviewer.

Round 2

Reviewer 1 Report

The manuscript has been improved regarding my concerns. 

Reviewer 2 Report

I want to thank the authors for their effort in responding adequately to the suggestions. This has made their work, already well done, even more interesting for the scientific community. I consider the work appropriate for publication.